# Empirical analysis of COVID-19 confirmed cases, hospitalizations, vaccination, and international travel across Belgian provinces in 2021

**Yessika Adelwin Natalia**[1]*, **Geert Molenberghs**[1,2], **Thomas Neyens**[1,2], **Niel Hens**[1,3], **Christel Faes**[1]

**1** I-BioStat, Data Science Institute, Hasselt University, Hasselt, Belgium, **2** I-Biostat, Leuven Biostatistics and Statistical Bioinformatics Centre, KU Leuven, Leuven, Belgium, **3** Centre for Health Economics Research and Modelling of Infectious Diseases (CHERMID), Vaccine and Infectious Disease Institute (VAXINFECTIO), University of Antwerp, Antwerp, Belgium

* yessikaadelwin.natalia@uhasselt.be

**Data availability statement:** All relevant data are within the manuscript and its Supporting Information files.

## Abstract

In the absence of definitive treatments or vaccines, the primary strategy to mitigate the COVID-19 pandemic relied on non-pharmaceutical interventions. By the end of 2020, COVID-19 vaccines had been developed and initiated for preventive purposes. To better understand the association between various mitigation measures and their impact on the pandemic, we analyzed the effect of vaccination coverage, international travel, traveler positivity rates, and the stringency of public health measures on the incidence of COVID-19 cases and hospitalizations at the provincial level in Belgium. We identified several important interactions among the covariates that influence the incidence of COVID-19 confirmed cases. Specifically, the best-fitting model (AIC = 965.658) revealed significant interactions between lagged vaccination coverage and the stringency index, as well as between incoming travel rates and positivity rates. Additionally, when modeling COVID-19 hospitalizations, a significant interaction was observed between the incoming travel rate and the stringency index. Model performance improved substantially when incorporating the incidence of confirmed cases as a covariate (AIC = 1061.516 vs. AIC = 432.708), while highlighting key interactions between confirmed cases and traveler positivity rates, as well as between lagged vaccination coverage and incoming travel rates. These findings underscore the intricate interplay between public health interventions, population immunity, and mobility patterns in shaping the course of the COVID-19 pandemic.

## Introduction

Four years have passed since the World Health Organization (WHO) declared the coronavirus disease 2019 (COVID-19), caused by severe acute respiratory syndrome coronavirus 2 (SARS-CoV-2), a pandemic [1]. By the time the pandemic status was officially lifted on May

**Funding:** TN gratefully acknowledges funding by the Internal Funds KU Leuven (project number 3M190682). The funders had no role in study design, data collection and analysis, decision to publish, or preparation of the manuscript.

**Competing interests:** The authors have declared that no competing interests exist.

5, 2023 [2], more than 765 million reported cases and almost seven million related deaths had been reported globally [3].

In the absence of a treatment or vaccine in 2020, the main mitigation strategy to prevent further escalation of the pandemic involved non-pharmaceutical interventions (NPIs), such as social distancing, use of face masks, and travel restrictions. Early in the pandemic, many countries implemented lockdowns to prevent further transmission or importation of SARS-CoV-2 [4]. Later, many countries adopted travel restrictions based on recent COVID-19 transmission in specific areas. In October 2020, the European Union introduced color-coded zones (green, orange, red, and gray) based on the risk of COVID-19 infections in a specific country to facilitate free movement while maintaining safety [5]. Stricter measures were implemented for travelers originating from high-risk zones. Several studies reported that travel restrictions, particularly on international travel, had a consistent effect on slowing down the spread of COVID-19 [6–8].

By the end of 2020, vaccines against COVID-19 had reached the market and could be added as a prophylactic measure. Europe started the vaccination campaign in December 2020, and per 23 January 2022, over 827 million vaccine doses had been administered in this region [9]. The first campaign focused on the older age population since COVID-19's severity increases with age and, consequently, hit the older age population hard in many periods [10]. As of May 1, 2022, 79% of the Belgian population had completed full primary COVID-19 vaccination, with 62% of those individuals also receiving a booster dose. Among adults aged 18 years and older, the full primary vaccination coverage was 88%, with 75% of this group having received a booster dose [11].

Many studies reported the effect of human mobility on the spread of COVID-19 [12–15] as well as the effect of COVID-19 vaccination on the incidence of COVID-19 cases or hospitalizations [16–18]. However, only a few studies reported the effects of human mobility, especially international travel, together with COVID-19 vaccination in the same analysis. Zou *et al.* simulated the influence of vaccination coverage and daily mobility among provinces on COVID-19's effective reproduction number during the Chinese-Spring-Festival travel rush in 2021 [19]. They concluded that vaccination decreases the reproductive number while high daily mobility yield an opposite effect. These results were based mainly on simulated data, thus it is important to evaluate the effect of these variables in a real-world setting.

Nguyen *et al.* reported the aggravating effect of international travel on the daily COVID-19 incidence in Belgium [20]. However, to our knowledge, no study has assessed the combined effect of COVID-19 vaccination and incoming international travel on confirmed cases and hospitalizations within a single analytical framework. Furthermore, it is crucial to account for additional factors that may influence disease dynamics, such as the positivity rate among incoming travelers and the public health mitigation measures implemented at different time points. To better understand the association between these factors and their impact on the COVID-19 pandemic, we conducted an analysis using publicly available data from Belgium. We focused our analysis on the year 2021, following the implementation of travel restrictions based on color-coded zones and the widespread roll-out of the COVID-19 vaccination campaign. By incorporating data on the positivity rate among incoming travelers and the mitigation measures in place during this time, we aim to provide a more comprehensive assessment of how international travel and vaccination jointly influenced COVID-19 transmission and healthcare burden.

## Materials and methods

### Study area

Belgium is divided into three regions: Flanders in the north, Wallonia in the south, and the Brussels-Capital Region in the center of Belgium. In 2021, the population of these regions was approximately 6.65 million in Flanders, 3.65 million in Wallonia, and 1.22 million in the Brussels-Capital Region. Both the Flemish and Walloon regions are further subdivided into five provinces, resulting in a total of 11 administrative units included in our analysis, as illustrated in Fig 1.

### Data

Data on daily confirmed COVID-19 cases and hospitalizations at the provincial level, along with weekly cumulative vaccinations at the municipality level, were made publicly available by Sciensano, the Belgian institute for public health [21].

Until May 23, 2022, all incoming travelers to Belgium, regardless of their mode of transportation, were required to complete a Passenger Locator Form (PLF) either prior to or upon arrival [22]. During this period, travelers arriving from high-risk zones were required to undergo mandatory testing on both day 1 and day 7 post-arrival. Sciensano documented the volume of incoming travelers and their test results in weekly epidemiological reports (available in Dutch, French, and German) [23]. Detailed weekly data on international arrivals and the day 1 post-arrival positivity rate at the provincial level are available up until early April 2022. To maintain consistency in our data and analysis, we restricted our study period to include only data from the year 2021.

To assess the general influence of travel restriction and other NPIs, we also retrieved the stringency index from Our World in Data [24]. A higher stringency index indicates more stringent policies at the country level during a given period [25]. The general population data in 2021 were obtained from StatBel, the Belgian National Statistics Institute [26].

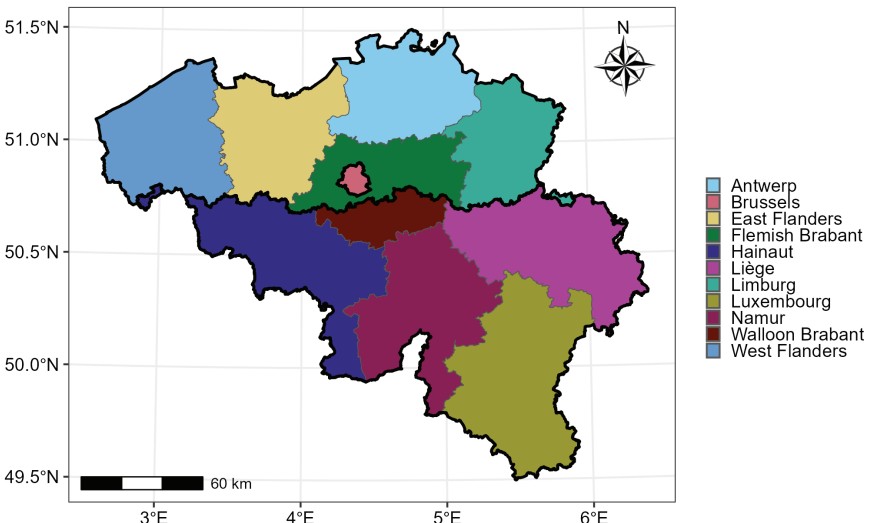

**Fig 1. Border of provinces in Belgium. Regional division is marked with black lines.** The map is adapted from https://statbel.fgov.be/en/open-data/statistical-sectors-2021 using R 4.4.1.

## Statistical methods

Let $Y_{ijk}$ be the COVID-19 incidence of confirmed cases per 100,000 individuals ($k = 1$) or the COVID-19 hospitalizations per 100,000 individuals ($k = 2$) at week $i = 1, \ldots, 52$ as defined by the International Standard ISO 8601, i.e., each week begins on Monday and week 1 is the first week with at least four days in the new year that contains the first Thursday, in province $j = 1, \ldots, 11$. Given the heterogeneity in outcomes across provinces, we used a logarithmic transformation of $Y_{ijk}$ and included province $j$ as a random effect. The general proposed model is given by:

$$\log Y_{ijk} = (\beta_{0k} + b_{0jk}) + \sum_{m=1}^{n} \beta_{mk} X_{mijk} + \varepsilon_{ijk}, \tag{1}$$

with $\varepsilon_{ijk} \sim N(0, \sigma_k^2)$, $b_{0jk} \sim N(0, \sigma_{Dk}^2)$, and $\varepsilon_{ijk}$, $b_{0jk}$ mutually independent. Here, $\beta_{0k}$ represents the outcome-specific fixed intercept and $b_{0jk}$ denotes the province-specific random intercept. The term $\beta_{mk}$ indicates the regression coefficient associated with each predictor, where $n$ represents the total number of effects, including any interaction terms, incorporated into the model.

In general, four main explanatory variables were considered in our analysis:

1. The weekly vaccination coverage per province defined as the cumulative population percentage that received full primary vaccination either with 1-dose or 2-doses vaccines. Taking into account that vaccination may have a delayed effect as immunity requires time to develop [27], we also explored the effect of vaccination coverage at lag $l$ week as a potential predictor. Given the duration of our study period and the potential for waning immunity [28,29], we used the lagged vaccination coverage up to six months prior to the current week. This variable is denoted as $\text{vfull}_{i-l,j}$ with $l = 0, 1, \ldots, 24$. On top of this, we also considered the non-linear effect of vaccination and a decrease in the marginal benefit at high vaccination coverage [30,31]; thus we included the inverse of vaccination coverage at lag $l$ week, denoted as $\frac{10}{\text{vfull}_{i-l,j}}$, in our model selection.

2. The weekly incoming travel rate ($\text{travel}_{ij}$) calculated as the number of incoming travelers (regardless of zone and mandate to get tested) to province $j$ at week $i$ per 100 inhabitants in that province (also denoted as percentage).

3. The weekly positivity rate among travelers from high-risk zones ($\text{pos}_{ij}$) calculated as the number of positive tests per 100 conducted tests (also denoted as percentage).

4. The median stringency index at the corresponding week $\text{SI}_i$.

It is important to note that COVID-19 hospitalizations are a direct consequence of COVID-19 infections. For this reason, we also evaluated an alternative model for hospitalizations that incorporates the incidence of confirmed cases in the same week as an explanatory variable. To avoid confusion, the logarithm of confirmed cases is denoted as $\log(\text{IC7})_{ij}$ when used as an explanatory variable in the model.

Considering different dynamics among these variables, we explored several fixed effects variations which include different interaction terms. To reduce the risk of overfitting, we constrained the model to include only two-way interaction terms. The most parsimonious model was selected based on the lowest Akaike information criterion (AIC) value.

As a sensitivity analysis, we selected a subset of our dataset from week 20 (May 17–23), a period in which vaccination coverage had reached at least 10% of the population in each province. Using this subset, we applied the same model selection procedure as that used for the full dataset to evaluate the robustness of our findings. All analyses were performed

using R 4.4.1 available from the Comprehensive R Archive Network (CRAN) (https://CRAN.R-project.org/).

## Results

### Exploratory data analysis

The weekly time trends of variables used in this study are presented in Fig 2.

Overall, multiple waves of confirmed cases were observed across all provinces (Fig 2A). A marked increase in the incidence of confirmed cases occurred between weeks 9 (March 1–7) and 17 (April 26–May 2), with the highest incidence recorded in Namur. A larger wave followed between weeks 41 (October 11–17) and 51 (December 20–26), peaking in West Flanders. An exception was seen in Brussels, where a modest increase was noted from week 25 (June 21–27), eventually developing into a more substantial wave by week 41.

During these periods, similar waves could be observed in hospitalizations (Fig 2B). Between weeks 9 and 17, a considerable increase in hospitalizations was noted in Brussels, Hainaut, and Namur. In Brussels, new hospitalizations increased again from week 25 until West Flanders reached higher incidence rates, particularly peaking around week 47 (November 22–28).

As shown in Fig 2C, the population percentage that received full primary vaccination increased considerably starting from week 17 (April 26–May 2). Every province reached 60% or higher by the end of week 33 (August 16–22), except for Brussels. By the end of 2021, West Flanders had the highest coverage of full primary vaccination (77.16%).

We observed a notable increase in the rate of incoming international travelers across all provinces from week 21 (May 24–30) onwards as illustrated in Fig 2D. However, there was a marked difference in Brussels where it maintained a consistently higher incoming travel rate throughout the year compared to other provinces.

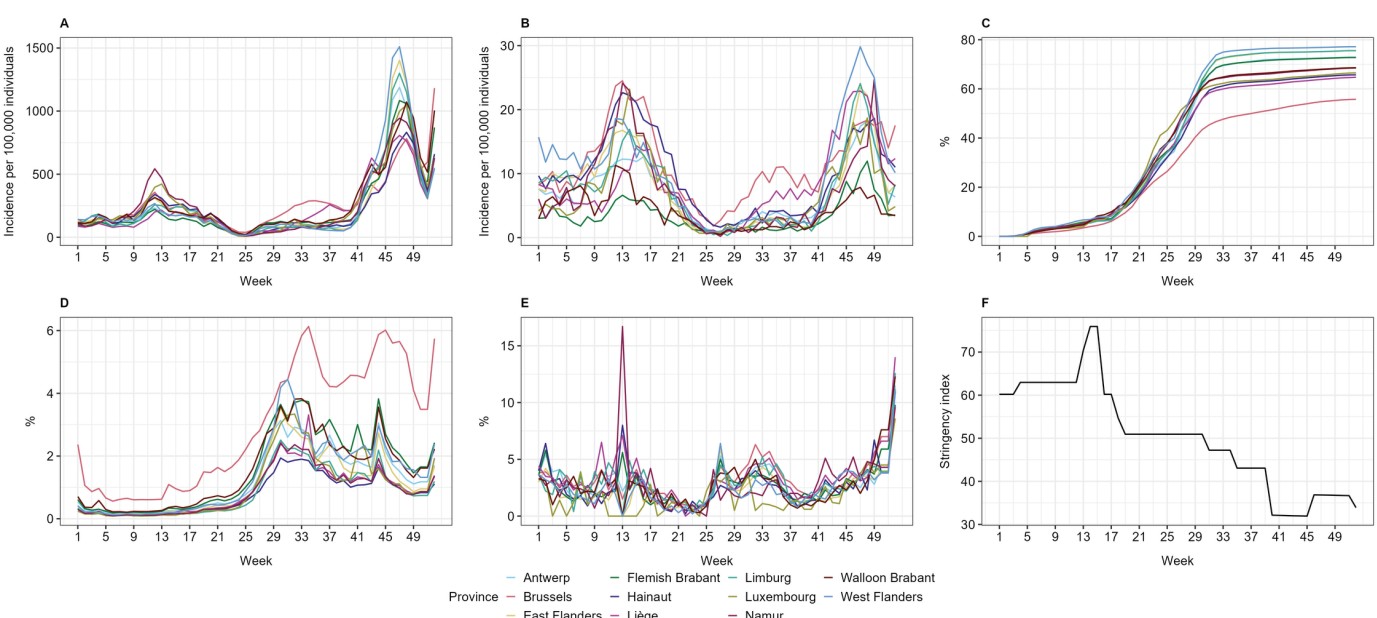

**Fig 2. Time trend of weekly (A) confirmed cases, (B) hospitalizations, (C) vaccination coverage, (D) incoming travel rate, (E) positivity rate among the incoming travelers, and (F) stringency index.**

The COVID-19 test positivity rate among travelers displayed variability over the year, with a general trend that appeared largely similar across the 11 provinces (Fig 2E). An exception could be observed in week 13 (March 29–April 4) when a pronounced peak in positivity rates among travelers who visited Namur was recorded.

The early months of 2021 were characterized by more stringent measures in Belgium, particularly between week 13 (March 29–April 4) and 15 (April 12–18, median stringency index = 75.93), as indicated in Fig 2F. When conditions improved into the summer months, some of these restrictions were relaxed, allowing for greater mobility and fewer constraints on travel. The median stringency index reached its lowest levels between week 41 (October 11–17) and 45 (November 8–14, median stringeency index = 43.98).

## Fitted linear mixed models

**COVID-19 confirmed cases.** The model selection process is outlined in S2 File, sheet S1 Table. Among the 2,900 candidate models evaluated for estimating $\log(Y_{ij1})$, one model achieved the lowest AIC value of 965.658, which is given by:

$$\log Y_{ij1} = (\beta_{01} + b_{0j1}) + \beta_{11}\text{vfull}_{i-14,j1} + \beta_{21}\text{travel}_{ij1} + \beta_{31}\text{pos}_{ij1} + \beta_{41}\text{SI}_{i1}$$
$$+ \beta_{51}\text{vfull}_{i-14,j1} \times \text{SI}_{i1} + \beta_{61}\text{travel} \times \text{pos}_{ij1} + \varepsilon_{ij1}, \tag{2}$$

with $\varepsilon_{ij1} \sim N(0, \sigma_1^2)$, $b_{0j1} \sim N(0, \sigma_{D1}^2)$, and $\varepsilon_{ij1}$, $b_{0j1}$ mutually independent. This model highlights the important effect of lagged vaccination coverage ($l = 14$), incoming travel rates, positivity rates, and the stringency index on the incidence of confirmed cases. Furthermore, significant two-way interactions were identified between lagged vaccination coverage and the stringency index, as well as between travel rates and positivity rates.

The complete parameter estimates of this model are shown in S2 File, sheet S2 Table. Using these estimates, we compared the observed incidence of confirmed cases with the 95% prediction intervals, as illustrated in Fig 3. The observed values consistently fall within the 95% prediction intervals, indicating a good overall fit of the model to the data. Notably, the width of the 95% prediction intervals increases starting from week 41 (October 11–17) in all provinces, suggesting greater uncertainty in the model's estimations during this period.

**COVID-19 hospitalizations.** Similar to the confirmed cases, a model with two interaction terms achieved the lowest AIC value of 1061.516 among the 2,900 candidate models. This model is denoted as:

$$\log Y_{ij2} = (\beta_{02} + b_{0j2}) + \beta_{12}\text{vfull}_{i-15,j2} + \beta_{22}\text{travel}_{ij2} + \beta_{32}\text{pos}_{ij2} + \beta_{42}\text{SI}_{i2}$$
$$+ \beta_{52}\text{vfull}_{i-15,j2} \times \text{SI}_{i2} + \beta_{62}\text{travel}_{ij2} \times \text{SI}_{i2} + \varepsilon_{ij2}, \tag{3}$$

with $\varepsilon_{ij2} \sim N(0, \sigma_2^2)$, $b_{0j2} \sim N(0, \sigma_{D2}^2)$, and $\varepsilon_{ij2}$, $b_{0j2}$ mutually independent. This model highlights again the important effect of lagged vaccination coverage ($l = 15$), incoming travel rates, positivity rates, and the stringency index on the incidence of hospitalizations. Significant two-way interactions were found between lagged vaccination coverage and the stringency index, as well as between travel rates and the stringency index.

All parameter estimates of this model are shown in S2 File, sheet S3 Table. As shown in Fig 4, the observed values consistently fall within the 95% prediction intervals. However, we observed a widening of the intervals during two distinct periods: prior to week 17 (April 26–-May 2) and after week 41 (October 11–-17), across all provinces.

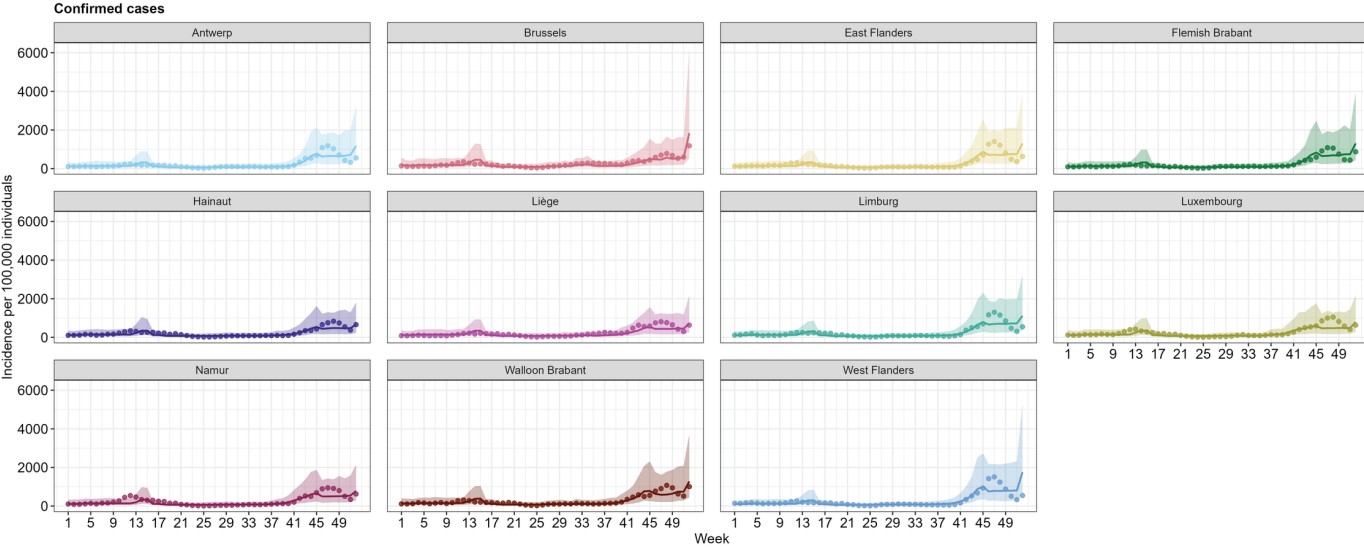

**Fig 3. Fitted values of the weekly incidence of confirmed cases based on Eq 2. Observed values are depicted by points around the prediction line.**

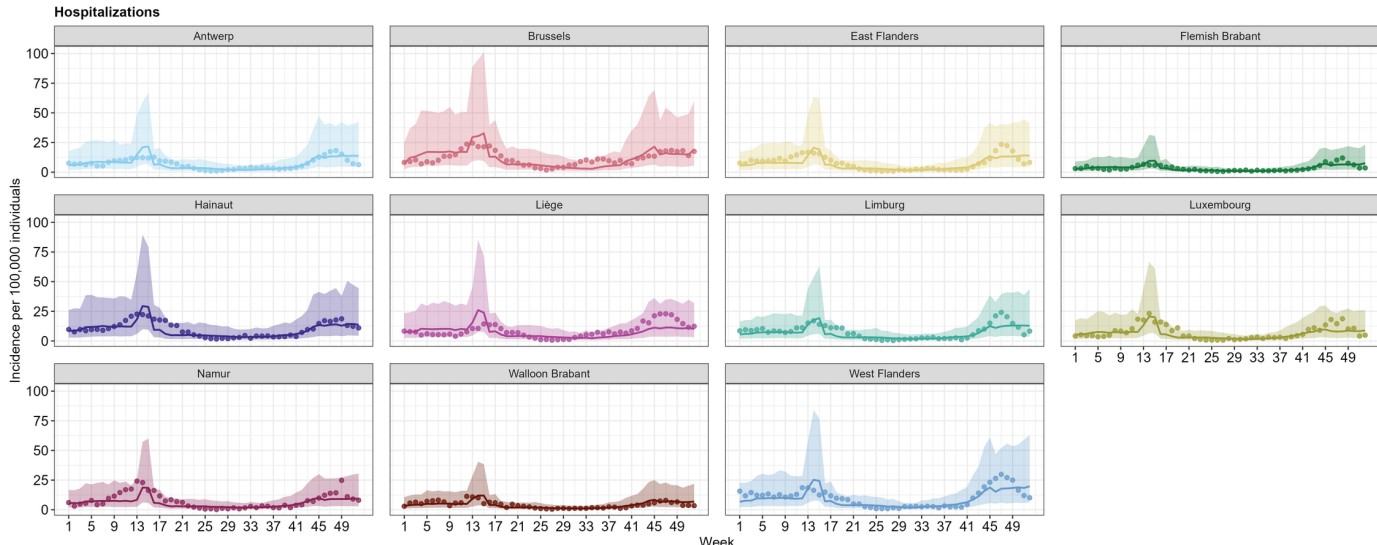

**Fig 4. Fitted values of the weekly incidence of confirmed cases based on Eq 3. Observed values are depicted by points around the prediction line.**

Incorporating the incidence of confirmed cases $\log(\text{IC7})_{ij}$ as an explanatory variable led to a considerable increase in the number of candidate models, with a total of 50,700 fitted models evaluated. Among these, the model achieving the lowest AIC value (432.708) is given by:

$$\log Y_{ij2} = (\beta_{02} + b_{0j2}) + \beta_{12}\text{vfull}_{i-3,j2} + \beta_{22}\text{travel}_{ij2} + \beta_{32}\text{pos}_{ij2} + \beta_{42}\text{SI}_{i2}$$
$$+ \beta_{52}\log(\text{IC7})_{ij2} + \beta_{62}\log(\text{IC7}) \times \text{pos}_{ij2} + \beta_{72}\text{vfull}_{i-3,j2} \times \text{travel}_{ij2}$$
$$+ \varepsilon_{ij2}, \tag{4}$$

with $\varepsilon_{ij2} \sim N(0, \sigma_2^2)$, $b_{0j2} \sim N(0, \sigma_{D2}^2)$, and $\varepsilon_{ij2}$, $b_{0j2}$ mutually independent. A detailed summary of the parameter estimates for this model is provided in S2 File, sheet S4 Table. This model demonstrated a substantial improvement over the previously reported model in Eq 3, as evidenced by its notably lower AIC value. The better performance was also reflected in the predictive accuracy, with predictions aligning more closely with observed data, as illustrated in Fig 5.

**Sensitivity analysis.** The model selection process using data from week 20 is summarized in S2 File, sheet S5 Table. Consistent with the results obtained from the full dataset, models incorporating vfull$_{i-l,j}$ as a predictor led to a substantially lower minimum AIC value compared to those including $\frac{10}{\text{vfull}_{i-l,j}}$, indicating a better model fit. An exception was observed, however, in the alternative model for hospitalizations that included $\log(\text{IC7})_{ij}$ as a predictor. In this case, the model using $\frac{10}{\text{vfull}_{i-l,j}}$ yielded the lowest AIC value, suggesting that the inverse transformation of vaccination coverage better captured the underlying dynamics of the COVID-19 hospitalizations for this shorter period. The formulations of the best-performing models are provided in Table 1.

The parameter estimates of these models are given in S2 File, sheet S6 Table. The predicted values and their corresponding intervals are presented in S1 Figure until S3 Figure.

## Discussion

In this study, we identified important effects of lagged vaccination coverage, the rate of incoming international travel, the positivity rate among travelers, and the stringency index on the dynamics of COVID-19 confirmed cases and hospitalizations. Furthermore, our results emphasize the potential interactions among these variables, suggesting that their combined impact may be different than merely the sum of their individual contributions.

Vaccination played a crucial role in reducing the incidence of COVID-19 cases and hospitalizations across various populations and geographic regions [32,33]. Consistent with these findings, our study demonstrates a significant association between vaccination coverage and

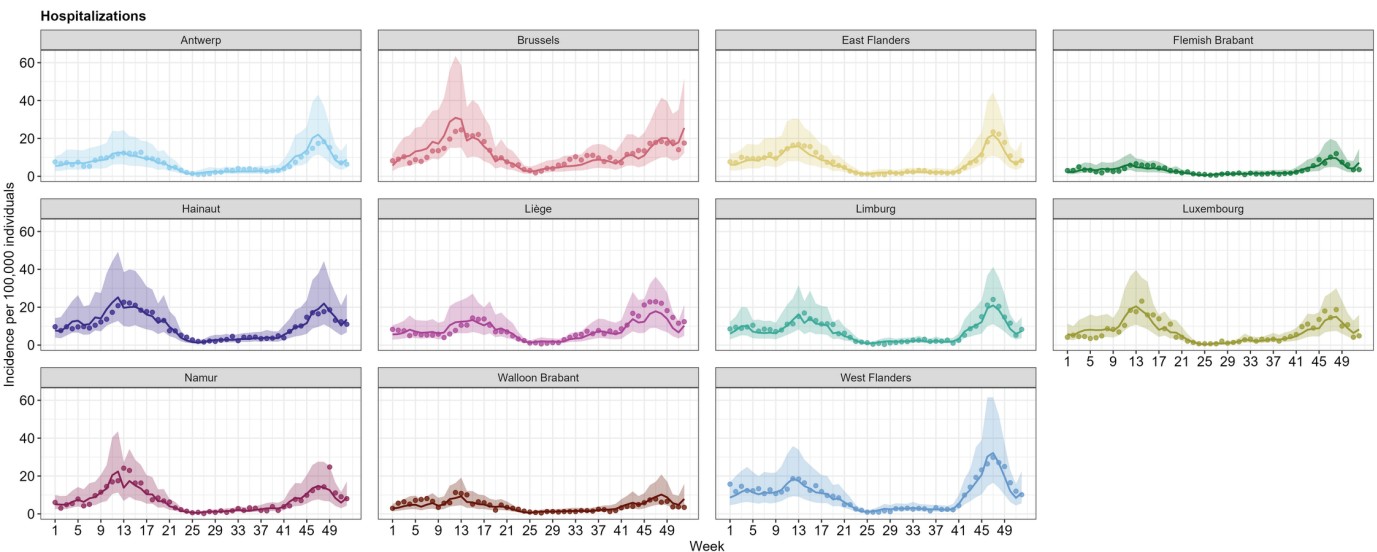

**Fig 5. Fitted values of the weekly incidence of confirmed cases based on Eq 4. Observed values are depicted by points around the prediction line.**

**Table 1. Optimal models to estimate the incidence of confirmed cases and hospitalizations using data from week 20.**

| Model notation | AIC |
|---|---|
| $\log Y_{ij1} = (\beta_{01} + b_{0j1}) + \beta_{11}\text{vfull}_{i-15,j1} + \beta_{21}\text{travel}_{ij1} + \beta_{31}\text{pos}_{ij1} + \beta_{41}\text{SI}_{i1} + \varepsilon_{ij1}$, with $\varepsilon_{ij1} \sim N(0, \sigma_1^2)$, $b_{0j1} \sim N(0, \sigma_{D1}^2)$, and $\varepsilon_{ij1}$, $b_{0j1}$ mutually independent. | 621.231 |
| $\log Y_{ij2} = (\beta_{02} + b_{0j2}) + \beta_{12}\text{vfull}_{i-23,j2} + \beta_{22}\text{travel}_{ij2} + \beta_{32}\text{pos}_{ij2} + \beta_{42}\text{SI}_{i2} + \beta_{52}\text{vfull}_{i-23,j2} \times \text{pos}_{ij2} + \beta_{62}\text{pos}_{ij2} \times \text{SI}_{i2} + \varepsilon_{ij2}$, with $\varepsilon_{ij2} \sim N(0, \sigma_2^2)$, $b_{0j2} \sim N(0, \sigma_{D2}^2)$, and $\varepsilon_{ij2}$, $b_{0j2}$ mutually independent. | 652.114 |
| $\log Y_{ij2} = (\beta_{02} + b_{0j2}) + \beta_{12}\frac{10}{\text{vfull}_{i-11,j2}} + \beta_{22}\text{travel}_{ij2} + \beta_{32}\text{pos}_{ij2} + \beta_{42}\text{SI}_{i2} + \beta_{52}\log(\text{IC7})_{ij2} + \beta_{62}\log(\text{IC7}) \times \text{pos}_{ij2} + \beta_{72}\log(\text{IC7}) \times \text{travel}_{ij2} + \beta_{82}\frac{10}{\text{vfull}_{i-11,j2}} \times \text{pos}_{ij2} + \varepsilon_{ij2}$, with $\varepsilon_{ij2} \sim N(0, \sigma_2^2)$, $b_{0j2} \sim N(0, \sigma_{D2}^2)$, and $\varepsilon_{ij2}$, $b_{0j2}$ mutually independent. | 312.963 |

AIC = Akaike information criterion.

COVID-19 incidences, as indicated by the negative coefficient of lagged vaccination coverage and/or its interaction term (S2 File, S2–S4 Tables). For most optimal models, we identified a similar lag *l* of approximately 14-15 weeks. The importance of considering lagged effects when evaluating the impact of vaccination on COVID-19 dynamics has been highlighted in previous research. For instance, Li et al. reported that the lag time for the protective effects of vaccination was approximately 40 days after the administration of the first dose of primary vaccination, with the potential for a rebound in epidemic intensity thereafter [27]. In contrast, Lokonon et al. observed a shorter lag of 15–20 days among hospitalized patients and those in intensive care units in Germany [34]. These discrepancies highlight the variability in lag times across different populations and study contexts. Notably, the shorter lag times reported by these studies compared to our findings may stem from differences in methodological approaches, particularly their use of daily data, which allows for finer temporal resolution and capturing more immediate effects.

While vaccination has been instrumental in providing individual immunity against COVID-19, it does not entirely eliminate the risk of infection [35]. Increased human mobility within a specific timeframe, particularly during periods of high SARS-CoV-2 circulation, elevates the likelihood of contact with an infected individual, thereby amplifying transmission risks. As shown in Fig 2, we observed a consistent trend starting from week 25 (June 21–27) where increases in incoming travel rates and positivity rates among travelers were accompanied by increases in COVID-19 confirmed cases and hospitalizations. This pattern reinforces findings from a previous study in Belgium, which demonstrated that international travel exacerbates COVID-19 incidence and suggested that restricting such travel could significantly mitigate epidemic growth [20]. Similarly, research from Ukraine reported a surge in COVID-19 cases during the summer of 2021, attributed to increased travel and tourism in the absence of travel restrictions [36]. In southern Taiwan, an outbreak in the summer of 2021 was linked to returning travelers from abroad, further underscoring the role of imported cases in local outbreaks [37].

The evidence collectively suggests that travel volume, particularly during peak periods like summer months, serves as a critical driver of COVID-19 transmission, especially when travelers originate from regions with high incidence rates [38]. However, such risks can be

mitigated through targeted interventions, such as frequent testing of travelers and commuters. This approach not only helps identify and isolate potential cases but also reduces the likelihood of widespread transmission, even in contexts where contact reduction policies are less stringent [39]. These findings underscore the importance of integrating mobility management strategies with public health measures to curtail the spread of COVID-19 while balancing the socio-economic demands of the population.

In addition to vaccination and international travel, several other factors evolved during the investigation period. At the beginning of 2021, the stringency index remained relatively high in response to the resurgence of infections during the fall of 2020 [40]. The restrictions were progressively eased starting from mid-April 2021 as part of a phased relaxation strategy (Fig 2F) following the nationwide vaccination roll out. The observed decline in stringency measures aligns with broader trends reported in European countries, where increased vaccination coverage facilitated policy shifts towards more lenient restrictions [41]. The strong association between vaccination coverage and the stringency index was further supported by our modeling results, which identified a significant interaction between these two variables (Eq 2 and 3, see also S2 File, S2 and S3 Tables). However, we also observed a widening of the 95% prediction intervals in the final weeks of 2021 as presented in Figs 3–5. This increased variability likely reflects the growing complexity of factors affecting COVID-19 incidences, including changes in dominant SARS-CoV-2 variants. In Belgium, the Alpha variant was predominant during the first half of 2021. However, the Delta variant began its ascent in May 2021, becoming dominant by July 1, 2021, and accounting for nearly 100% of cases by August 1, 2021. Subsequently, the Omicron variant emerged and began circulating in December 2021. These shifts in variants are critical, as each variant exhibits unique transmissibility and immune evasion characteristics [42,43], which introduced additional complexity to the epidemiological landscape, influencing transmission dynamics and the effectiveness of NPIs.

We found a substantial improvement in model performance when incorporating COVID-19 confirmed cases as a covariate to estimate COVID-19 hospitalizations as indicated by the AIC value (1,061.516 for model in Eq 3 and 432.708 for model in Eq 4). This result highlights the critical role of confirmed cases in explaining hospitalization trends, as they represent a proximal indicator of disease burden within the population. Including this covariate not only enhanced the predictive accuracy of the model but also enabled a more comprehensive representation of interactions with other variables, such as vaccination coverage and non-pharmaceutical interventions. These interactions are crucial for capturing the multifaceted dynamics influencing hospitalization trends.

Our sensitivity analysis further revealed that using inverse lagged vaccination coverage as a predictor has a potential to improve model's ability to evaluate COVID-19 hospitalizations. This transformation likely reflects the delayed and non-linear relationship between vaccination efforts and their protective effects at the population level. Specifically, it accounts for diminishing returns in protection as vaccination coverage approaches saturation, variations in immunity due to waning effectiveness, and temporal lags in immunity development post-vaccination. While non-linear mathematical modeling frameworks have frequently addressed such effects [44–46], incorporating these transformations into linear models, which are often used in epidemiological research and public health decision-making, remains a valuable approach to bridge the gap between simplicity and the nuanced nature of epidemiological dynamics.

Some limitations in our analyses should be mentioned. The models employed are statistical and descriptive in nature, designed primarily for interpolation rather than extrapolation. Their application to predictions beyond the observed data ranges of the predictor variables should be approached with caution. When the emphasis is on mobility patterns across

different areas such as regions or provinces, a meta-population approach might be of relevance [8,47].

On top of this, hospitalization data from smaller provinces over certain time periods exhibited significant variability, introducing additional noise into the models. Regional and provincial differences in socio-economic factors, cultural norms, and contact patterns can further complicate the analysis [48,49]. Evidently, this is captured to some extent in the various travel rates. Nevertheless, variations in contact behaviors—both domestically and at travel destinations—are likely contributors to the observed heterogeneity but remain unaccounted for in this study.

Further, it should be noted that the stringency index data are only available at the country level, therefore, they do not capture variations in policy stringency at lower administrative levels. The lack of localized data on stringency measures and SARS-CoV-2 variants limited our ability to comprehensively evaluate the impact of non-pharmaceutical interventions (NPIs) and variant-specific dynamics at the provincial level. Additionally, we assume a similar evolution of SARS-CoV-2 variants for the whole country, which may not hold true when considering finer spatial or temporal resolutions. Future studies incorporating more granular data are essential to better disentangle these effects and refine our understanding of the interplay between these variables.

## Conclusion

Our findings highlight the complex interplay between public health interventions, population immunity, and mobility patterns in shaping the COVID-19 pandemic. We identified different interaction patterns among the covariates that influence the incidence of COVID-19 confirmed cases and hospitalizations. The best-fitting model for estimating confirmed cases (AIC = 965.658) revealed significant interactions between lagged vaccination coverage and the stringency index, as well as between incoming travel rates and positivity rates. When modeling COVID-19 hospitalizations, a significant interaction was observed between the incoming travel rate and the stringency index. Model performance improved substantially upon incorporating the incidence of confirmed cases as a covariate (AIC = 1,061.516 vs. AIC = 432.708), while highlighting key interactions between confirmed cases and traveler positivity rates, as well as between lagged vaccination coverage and incoming travel rates. For most optimal models, we identified a similar lag $l$ of approximately 14-15 weeks. However, this lag duration should be interpreted with caution, as each model includes a different set of interaction terms. These insights contribute to a deeper understanding of the factors driving COVID-19 transmission and healthcare burden, offering valuable guidance for optimizing policy responses in future epidemic scenarios.

## Supporting information

**S1 File. Full dataset for fitting the linear mixed models.**
(CSV)

**S2 File. Model selection process and parameter estimates of the final models (S1–S6 Tables).**
(XLSX)

**S1 Figure. Fitted values of the weekly incidence of confirmed cases using data from week 20. Observed values are depicted by points around the prediction line.**
(TIF)

**S2 Figure. Fitted values of the weekly incidence of hospitalizations (without log(IC7)$_{ij}$) using data from week 20. Observed values are depicted by points around the prediction line.**
(TIF)

**S3 Figure. Fitted values of the weekly incidence of hospitalizations (with log(IC7)$_{ij}$) using data from week 20. Observed values are depicted by points around the prediction line.**
(TIF)

## Author contributions

**Conceptualization:** Yessika Adelwin Natalia, Geert Molenberghs, Thomas Neyens, Niel Hens, Christel Faes.

**Data curation:** Yessika Adelwin Natalia, Geert Molenberghs.

**Formal analysis:** Yessika Adelwin Natalia.

**Funding acquisition:** Thomas Neyens.

**Investigation:** Geert Molenberghs.

**Methodology:** Geert Molenberghs, Thomas Neyens, Niel Hens, Christel Faes.

**Software:** Yessika Adelwin Natalia.

**Supervision:** Geert Molenberghs, Thomas Neyens, Niel Hens, Christel Faes.

**Visualization:** Yessika Adelwin Natalia.

**Writing – original draft:** Yessika Adelwin Natalia.

**Writing – review & editing:** Geert Molenberghs, Thomas Neyens, Niel Hens, Christel Faes.

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
