## [Decision Letter · Decision Letter 0]

17 Oct 2024

PONE-D-24-10186Empirical association between COVID-19 confirmed cases, hospitalization, vaccination, and travel: Comparison among different regions and provinces in Belgium between April-July 2021PLOS ONE

Dear Dr. Natalia,

Thank you for submitting your manuscript to PLOS ONE. After careful consideration, we feel that it has merit but does not fully meet PLOS ONE’s publication criteria as it currently stands. Therefore, we invite you to submit a revised version of the manuscript that addresses the points raised during the review process.

The manuscript has been evaluated by two reviewers, and their comments are available below. I am grateful to both of the reviewers for their detailed assessments of your manuscript. Both reviewers raised a number of major and minor concerns. They feel the manuscript should outline a clearly-defined research question, and they request improvements to the reporting of methodological and statistical aspects of the study. Please see the two attached files for the full reviews.

Could you please carefully revise the manuscript to address all comments raised?

We look forward to receiving your revised manuscript.

Kind regards,

Steve Zimmerman, PhD

Senior Editor, PLOS ONE

Journal Requirements:

2. We note that [Figure 1] in your submission contain [map/satellite] images which may be copyrighted. All PLOS content is published under the Creative Commons Attribution License (CC BY 4.0), which means that the manuscript, images, and Supporting Information files will be freely available online, and any third party is permitted to access, download, copy, distribute, and use these materials in any way, even commercially, with proper attribution. For these reasons, we cannot publish previously copyrighted maps or satellite images created using proprietary data, such as Google software (Google Maps, Street View, and Earth). For more information, see our copyright guidelines: http://journals.plos.org/plosone/s/licenses-and-copyright.

Reviewers' comments:

Reviewer's Responses to Questions

**Comments to the Author**

1. Is the manuscript technically sound, and do the data support the conclusions?

Reviewer #1: Partly

Reviewer #2: Partly

2. Has the statistical analysis been performed appropriately and rigorously? 

Reviewer #1: I Don't Know

Reviewer #2: Yes

3. Have the authors made all data underlying the findings in their manuscript fully available?

Reviewer #1: Yes

Reviewer #2: Yes

4. Is the manuscript presented in an intelligible fashion and written in standard English?

Reviewer #1: No

Reviewer #2: Yes

5. Review Comments to the Author

Reviewer #1: see attachment for comments

Reviewer #2: The article ``Empirical association between COVID-19 confirmed cases,

hospitalization, vaccination, and travel [...]'' by Yessika Adelwin Natalia et al. presents a statistical analysis between the mentioned explanatory and to be explained variables. One should maybe add ``international'' or better ``air'' in the title in front of travel, as other travel is not considered.

Please see attached pdf for detailed review comments.

6. PLOS authors have the option to publish the peer review history of their article (what does this mean?). If published, this will include your full peer review and any attached files.

Reviewer #1: No

Reviewer #2: No

---

## [Author Response · Author response to Decision Letter 1]

21 Jan 2025

We want to thank the editor and reviewers for their critical assessment of our manuscript and their constructive comments. We have addressed all of them and modified the paper accordingly. Please find our detailed response in a separate document.

---

## [Decision Letter · Decision Letter 1]

20 Feb 2025

PONE-D-24-10186R1Empirical analysis of COVID-19 confirmed cases, hospitalizations, vaccination, and international travel across Belgian provinces in 2021PLOS ONE

Dear Dr. Natalia,

Thank you for submitting your manuscript to PLOS ONE. After careful consideration, we feel that it has merit but does not fully meet PLOS ONE’s publication criteria as it currently stands. Therefore, we invite you to submit a revised version of the manuscript that addresses the points raised during the review process.

We look forward to receiving your revised manuscript.

Kind regards,

Dr. Hani Amir Aouissi

Academic Editor

PLOS ONE

Journal Requirements:

Reviewers' comments:

Reviewer's Responses to Questions

**Comments to the Author**

1. If the authors have adequately addressed your comments raised in a previous round of review and you feel that this manuscript is now acceptable for publication, you may indicate that here to bypass the “Comments to the Author” section, enter your conflict of interest statement in the “Confidential to Editor” section, and submit your "Accept" recommendation.

Reviewer #2: All comments have been addressed

Reviewer #3: (No Response)

2. Is the manuscript technically sound, and do the data support the conclusions?

Reviewer #2: Yes

Reviewer #3: Yes

3. Has the statistical analysis been performed appropriately and rigorously? 

Reviewer #2: Yes

Reviewer #3: Yes

4. Have the authors made all data underlying the findings in their manuscript fully available?

Reviewer #2: Yes

Reviewer #3: Yes

5. Is the manuscript presented in an intelligible fashion and written in standard English?

Reviewer #2: Yes

Reviewer #3: Yes

6. Review Comments to the Author

Reviewer #2: (No Response)

Reviewer #3: It was a pleasure for me to read this paper:

Title : « Empirical analysis of COVID-19 confirmed cases, hospitalizations, vaccination, and international travel across Belgian provinces in 2021 »,

Reference : PONE-D-24-10186R1

Journal : PLOS ONE.

After reviewing it, I think the paper is very qualitative. In addition, given the fact that this is a revised version, the authors previously improved the overall level of their manuscript, this is why I will just give some minor comments in order to help to optimize the quality, if you don’t mind here are some comments below :

1. In the abstract section : «…The analysis revealed crucial effects of these factors on the dynamics of COVID-19 cases and hospitalizations. Notably, the results suggest potential interactions among these variables, indicating that their combined effects may differ from the sum of their individual contributions… ». I think you have to support your statements by adding some results (numbers).

2. Lines 38-45, I appreciated the justification of your study. Nevertheless, I think you have to finish your abstract by clearly specifying the aim of your study, It may help the readers to better understand what you’ve done and why.

3. Fig 1. Normally the scale should be put in the lower part of the map . In addition, please add the North direction and the source (eg. Authors) if you have drawn it.

4. I suggest dividing your materials & methods section into 3 subsections: 1. Study Area 2. Data 3. Statistical Methods. You can include a revised version of your map and adding some information about Belgium in general.

5. Isn't there more recent data than 2021? I think we should try to justify this by explaining this in the manuscript ?

6.Line 171, you mean AIC= 432.708 ? If yes, please specify

7. I think there is a lack of references in the discussion section, I recommend to discuss the findings of the following papers, or to read them (if not the case) in addition of other manuscripts, here are only examples:

Doi: 10.1038/s41598-022-05498-z / Doi: 10.1016/j.ijbiomac.2022.01.118 / Doi: 10.3390/vaccines10111781 / Doi: 10.1080/01559982.2022.2045418

8. Same comment as the abstract section, please consider improving your discussion by emphasizing some quantitative results to support some statements.

9. Line 258-291, please avoid using first, second,…etc while describing your limitations.

10.Conclusion sections looks a little weak, please try to extend it a bit.

11. The references are well chosen and the majority are recent, it was a good job. Again, I just suggest to increase their number in order to better support your work.

7. PLOS authors have the option to publish the peer review history of their article (what does this mean?). If published, this will include your full peer review and any attached files.

Reviewer #2: No

Reviewer #3: No

---

## [Author Response · Author response to Decision Letter 2]

6 Mar 2025

We thank the reviewers for their time and effort in reviewing our revised manuscript and for the nice appraisal of our work. We have addressed all comments in a separate document.

---

## [Decision Letter · Decision Letter 2]

16 Mar 2025

Empirical analysis of COVID-19 confirmed cases, hospitalizations, vaccination, and international travel across Belgian provinces in 2021

PONE-D-24-10186R2

Dear Dr. Natalia,

We’re pleased to inform you that your manuscript has been judged scientifically suitable for publication and will be formally accepted for publication once it meets all outstanding technical requirements.

Kind regards,

Hani Amir Aouissi

Academic Editor

PLOS ONE

Additional Editor Comments (optional):

Reviewers' comments:

Reviewer's Responses to Questions

**Comments to the Author**

1. If the authors have adequately addressed your comments raised in a previous round of review and you feel that this manuscript is now acceptable for publication, you may indicate that here to bypass the “Comments to the Author” section, enter your conflict of interest statement in the “Confidential to Editor” section, and submit your "Accept" recommendation.

Reviewer #3: All comments have been addressed

2. Is the manuscript technically sound, and do the data support the conclusions?

Reviewer #3: Yes

3. Has the statistical analysis been performed appropriately and rigorously? 

Reviewer #3: Yes

4. Have the authors made all data underlying the findings in their manuscript fully available?

Reviewer #3: Yes

5. Is the manuscript presented in an intelligible fashion and written in standard English?

Reviewer #3: Yes

6. Review Comments to the Author

Reviewer #3: The revised version of the manuscript titled « Empirical analysis of COVID-19 confirmed cases,

hospitalizations, vaccination, and international travel across Belgian provinces in 2021 », Ref « PONE-D-

24-10186R2 » was well handled.

The authors responded to almost all my comments and suggesstions and made appropriate qualitative

revisions in the main text, I think that it was a very good job and that the paper is now suitable for

publication in my humble opinion.

Thank you again for the opportunity to read this article.

7. PLOS authors have the option to publish the peer review history of their article (what does this mean?). If published, this will include your full peer review and any attached files.

Reviewer #3: No

---

## [Editor Report · Acceptance letter]

PONE-D-24-10186R2

PLOS ONE

Dear Dr. Natalia,

I'm pleased to inform you that your manuscript has been deemed suitable for publication in PLOS ONE. Congratulations! Your manuscript is now being handed over to our production team.

Kind regards,

on behalf of

Dr. Hani Amir Aouissi

Academic Editor

PLOS ONE